# The Expression and Epigenetic Characteristics of the HSF2 Gene in Cattle-Yak and the Correlation with Its Male Sterility

**DOI:** 10.3390/ani14101410

**Published:** 2024-05-08

**Authors:** Qinhui Yang, Yumian Xie, Bangting Pan, Yuying Cheng, Yanjin Zhu, Xixi Fei, Xupeng Li, Jun Yu, Zhuo Chen, Jian Li, Xianrong Xiong

**Affiliations:** 1Key Laboratory for Animal Science of National Ethnic Affairs Commission, Southwest Minzu University, Chengdu 610041, China; qinhui_yang@163.com (Q.Y.); yumian_xie@163.com (Y.X.); yuying156960@163.com (Y.C.); zhuyj199434@163.com (Y.Z.); 18215163537@163.com (X.L.); zcheni4u@163.com (Z.C.); jianli_1967@163.com (J.L.); 2Key Laboratory of Qinghai-Tibetan Plateau Animal Genetic Resource Reservation and Exploitation of Ministry of Education, Southwest Minzu University, Chengdu 610041, China; pbtywzq@163.com (B.P.); feixixi_special@163.com (X.F.); yujun2000813@163.com (J.Y.)

**Keywords:** *HSF2*, spermatogenesis, cattle-yak, male sterility, methylation

## Abstract

**Simple Summary:**

Abnormal expression of the *HSF2* gene was shown to be associated with male sterility. Here, we found that *HSF2* was highly expressed in the testes of the cattle-yak, especially in adult cattle-yak. However, the expression of *HSF2* was significantly lower in the cattle-yak compared to cattle and yak. In addition, the methylation level of the promoter region was significantly higher in cattle-yak compared to yak. We therefore suggest that male cattle-yak sterility may be associated with methylation of the *HSF2* gene in testes tissues. This study provides theoretical support for further understanding of male cattle-yak sterility.

**Abstract:**

Aberrant expression of the heat shock proteins and factors was revealed to be closely associated with male reproduction. Heat shock factor 2 (HSF2) is a transcription factor that is involved in the regulation of diverse developmental pathways. However, the role and the corresponding molecular mechanism of *HSF2* in male cattle-yak sterility are still poorly understood. Therefore, the aim of this study was to obtain the sequence and the biological information of the cattle-yak *HSF2* gene and to investigate the spatiotemporal expression profiles of the locus during the development of cattle-yak testes. Additionally, the differential expression was analyzed between the cattle-yak and the yak, and the methylation of corresponding promoter regions was compared. Our results showed an additional 54 bp fragment and a missense mutation (lysine to glutamic acid) were presented in the cattle-yak *HSF2* gene, which correlated with enriched expression in testicular tissue. In addition, the expression of the *HSF2* gene showed dynamic changes during the growth of the testes, reaching a peak in adulthood. The IHC indicated that HSF2 protein was primarily located in spermatocytes (PS), spermatogonia (SP), and Sertoli cells (SC) in cattle-yak testes, compared with the corresponding cells of cattle and the yak. Furthermore, bisulfite-sequencing PCR (BSP) revealed that the methylated CpG sites in the promoter region of the cattle-yak HSF2 were more numerous than in the yak counterpart, which suggests hypermethylation of this region in the cattle-yak. Taken together, the low expression abundance and hypermethylation of *HSF2* may underpin the obstruction of spermatogenesis, which leads to male cattle-yak infertility. Our study provided a basic guideline for the *HSF2* gene in male reproduction and a new insight into the mechanisms of male cattle-yak sterility.

## 1. Introduction

The Yak (*Bos grunniens*), as a representative species of the Qinghai-Tibetan Plateau, is one of the bovine subfamilies known to thrive in the alpine pastures and the vicinities. Yak can make full use of the pasture resources of alpine meadows with strong adaptability to the local ecological conditions and harsh environments (low temperature, low oxygen content, strong UV light) [1,2]. As one of the livestock that was able to subsist and breed in this area, the yak has become the primary livelihood for the local herdsmen, providing meat, milk, and fur [3,4]. Because of the extreme climate and grass insufficiency, the fertility and productivity of yak are relatively lower than those of their plain cattle counterparts, which has become a major bottleneck for the development of local economy and animal husbandry [5]. To improve this prominent issue and enhance the productive performance of yak, lots of excellent gene resources, such as plain cow and cattle, were introduced to hybridize with yak to ameliorate the production traits, and this F1 generation was generally called the cattle-yak [5]. There is predominant heterosis in the cattle-yak, including identical adaptability to the harsh environment in plateaus as the yak, as well as high growth performance, and milk production performance [6]. Thus, the breeding of cattle-yak is thought to be one of the most effective ways to solve the dilemma of yak cultivation industrialization and is also beneficial for protecting the ecology of that region. Unfortunately, due to the blocking of spermatogenesis, the male cattle-yak is infertile, which hinders the maximum utilization of their heterosis.

Hybrid sterility is a common genetic effect that prevents the full exploitation of hybrid advantage in mammals [7], which has a genetic effect in that males tend to become infertile earlier than females in XY sex-determined species [8]. However, the causes of male sterility are various and complex, ranging from dysfunctional genetics to uncomfortable environmental factors. A recent study has underscored that aberrant gene expression can impair sperm production, thereby affecting fertility [9]. Moreover, testicular dysfunction, chromosomal abnormalities, and genetic mutations played direct or indirect roles in male sterility [10,11]. In addition, changes in the environment, such as temperature, oxygen concentration, and ultraviolet light may lead to cellular oxidative stress or epigenetic dysregulation, which were implicated in male infertility [12,13]. However, the underlying regulatory mechanism of hybrid male sterility has not yet been functionally elucidated, and the strategy for correcting this error remains unavailable.

Heat shock proteins (HSPs) and heat shock factors (HSFs) are ubiquitously expressed in mammalian tissues and organisms [14], which are universally acknowledged for the protective response of cells or tissue to heat stress in vivo or in vitro via HSFs [15]. Yet, little is known about the specific function of HSFs in mammalian gametogenesis. Spermatogenic cells are extremely sensitive to heat shock as the external environmental temperature increases or decreases. A previous study showed that elevated testicular temperature has a negative effect on male reproduction, including markedly disrupting spermatogenesis and impairing sperm quality, which can increase the risk of male infertility and compromise semen quality [16]. Interestingly, a significant downregulation was observed in sperm concentration and quantity when the human scrotum was tentatively exposed to a temperature in excess of body temperature [17,18]. Growing evidence suggests that *HSF1* and *HSF2* are the main members of the HSF family, which can maintain normal spermatogenesis in male animals [19,20]. In mice, *HSF2* was found to be primarily expressed in pachytene-stage spermatocytes and round spermatids in vivo [21]. With the knockdown or absence of the expression of *HSF2* in mice, the testicle size was remarkably reduced, accompanied by a decreased sperm count and vacuolization of the seminiferous tubules, which affected male reproduction to a certain extent [22]. Unexpectedly, knockout of both *HSF1* and *HSF2* at the same time can lead to male mice infertility with no spermatogenesis in the seminiferous tubule [22]. In female mice, it was also found that deficiency of the *HSF2* gene could result in a marked decrease in fertility and reproduction, which is accompanied by abnormal oocyte and embryonic development [23,24]. Importantly, the accumulating evidence suggests that a significant number of coding mutations in the exons of *HSF2* were found in idiopathic azoospermia (IA) patients, indicating *HSF2* played a crucial role in human spermatogenesis via regulating the cell cycle and apoptosis [25,26]. This study also pointed out that most of these mutations were heterozygous and usually caused the dysfunction of *HFS*2 [26]. According to the perspective of structural biology, it has been found that HSF2 protein is a short-lived protein that is less stable compared to the HSF1 protein and may degrade within one hour due to heat stress after ubiquitination by the ubiquitin E3 ligase anaphase-promoting complex/cyclosome (APC/C) [27,28]. Therefore, our logic suggests that male sterility in the cattle-yak may be attributed to the abnormal expression and epigenetic modification of HSFs.

In the present study, we first cloned the *HSF2* gene in the cattle-yak and analyzed the molecular and biological characteristics using bioinformatics. The expression profile of *HSF2* mRNA in diverse tissues of the cattle-yak was detected, as was the spatiotemporal expression pattern during different developmental stages. Moreover, immunohistochemistry (IHC) was utilized to identify the localization of the HSF2 protein in the testes of the cattle-yak and its counterparts. The methylation status of the promotor region in *HSF2* was explored by bisulfite-sequencing PCR (BSP) to further investigate the relationship between epigenetic modification and the expression abundance of the *HSF2* gene in the cattle-yak. Our study could advance the association between the expression profile of *HSF2* and the male sterility of the cattle-yak, and accelerate the analysis of the molecular underlying mechanism of male infertility in the cattle-yak.

## 2. Materials and Methods

### 2.1. Ethics Statement

All animals involved were approved by the Academic Committee of Southwest Minzu University (Chengdu, Sichuan, China). In addition, all experiments were conducted in accordance with the National Research Council’s guidelines for animal welfare and ethics.

### 2.2. Sample Collection

Each tissue sample, including heart, liver, spleen, lungs, kidneys, intestine, muscles, and testes, was captured from three adult yak and cattle-yak, respectively in a pasture in Hongyuan County, Sichuan Province (altitude 3500 m; longitude 102°44′ E; latitude 32°41′ N). In addition, the same tissue samples of adult cattle were obtained randomly from three healthy cattle in Qingbaijiang District, Sichuan Province. Herein, testes tissue samples from fetal (6 months old), juvenile (1 year old), and adult cattle-yak (3 years old) were obtained with three biological replicates, respectively. All samples were cleaned with autoclaved PBS, and then parts of them were put into cryopreservation tubes and brought back to the laboratory in liquid nitrogen tanks for subsequent experiments. The other parts of the tissues were fixed with 4% paraformaldehyde for IHC detection. Remarkably, all the animals on this ranch lived in the same environment and enjoyed comparable nutritional conditions. In addition, the selected samples were all from healthy individuals with a similar age and body size. In this study, all the samples were collected in September, and they are in a similar physiological state. Each evaluated group involved three healthy individuals who were of the same age, had a similar physiological status and body size, and comprised 4 technical replicates for each trial.

### 2.3. Gene Cloning

Total RNA was extracted with TRIzol (Vazyme, Nanjing, China) under the instruction of the manufacturer, followed by the detection of concentration and purity of total RNA by a UV spectrophotometer (Biospec-nano, Tokyo, Japan). The synthesis of cDNA was performed according to the instructions of the Revert Aid First Strand cDNA Synthesis Kit (Thermo Scientific, Waltham, MA, USA). Then, the cDNA samples were preserved at 80 °C for subsequent experiments. The specific primers (shown in Table 1) were designed by NCBI Primer-BLAST (NCBI, Bethesda, Rockville, MD, USA) based on the predicted sequence of Bos mutus (Accession number: XM_005890111.1) and synthesized at Sangon Biotech Co. (Shanghai, China). The PCR reaction system (25 µL) was selected, including upstream primer 1 µL, downstream primer 1 µL, 2× Rapid Taq Master Mix 12.5 µL, cDNA 2 µL, and ddH_2_O 8.5 µL. The PCR reaction procedure was pre-denaturalized at 93 °C for 4 min, followed by 38 cycles containing 93 °C for 12 s, 65 °C (annealing temperature) for 12 s, and 72 °C for 28 s. Then, it was extended at 72 °C for 6 min. The PCR products were detected by 1.5% agarose gel electrophoresis before being sent to Sangon Biotech Co. (Shanghai, China) for sequencing.

### 2.4. Bioinformatics Analysis

The *HSF2* gene sequences were analyzed by the PRF Reader Program of NCBI (https://www.ncbi.nlm.nih.gov/orffinder/ (accessed on 20 October 2023)). Amino acid sequences were predicted by NCBI (https://www.ncbi.nlm.nih.gov/ (accessed on 20 October 2023)), and the evolutionary tree was generated by MEGA7 (Oxford University, Oxford, UK) analysis. The physicochemical properties of the proteins were then inspected by the online software Expasy Protparam (https://web.expasy.org/protparam/ (accessed on 20 October 2023)), and the signal peptides were inspected by the online software SignalP 5.0 (https://services.healthtech.dtu.dk/services/SignalP-5.0/ (accessed on 20 October 2023)). The domain of proteins was anticipated by the online software InterPro 99.0 (http://www.ebi.ac.uk/interpro/ (accessed on 20 October 2023)). In order to predict the secondary structure, tertiary structure, and interaction proteins of HSF2 protein, online tools SOPMA (https://npsa-prabi.ibcp.fr/ (accessed on 20 October 2023)), SWISS-MODEL (https://swissmodel.expasy.org/ (accessed on 20 October 2023)), and STRING11.0 (https://string-db.org/ (accessed on 20 October 2023)) were used, respectively.

### 2.5. Real-Time Quantitative PCR (RT-qPCR)

The related primers for RT-qPCR were designed with online software and shown in Table 1. The relative mRNA expression of the *HSF2* gene was normalized to the mean abundance of the endogenous control gene beta-actin (*β-actin*). The amplification system (10 µL) included 5 µL 2× NovoStart^®®^ SYBR qPCR SuperMix Plus (Novoprotein, Suzhou, China), 1.5 µL cDNA, 0.4 µL forward and reverse primers, and RNase-free water 2.7 µL. Next, the CFX96TM system (Bio-Rad, Hercules, CA, USA) was used for detecting the expression of *HSF2*, as in our previous study [29]. The relative expression level against *β-actin* was analyzed with the help of the 2^−ΔΔCt^ [30] method. The detections were repeated at least three times. In order to make the analysis more convenient, the expression of the spleen was treated as 1 and set as the control group in tissue differential expression analysis. Similarly, the fetal cattle-yak group was defined as 1 when comparing *HSF2* expression in different species (cattle, yak, and cattle-yak).

### 2.6. Protein Isolation and Western Blotting

The total proteins were extracted under the instructions of the Total Protein Extraction Kit (No. BC3710, Solarbio, Beijing, China). The protein concentration and purity were determined by a microplate reader using the BCA Protein Assay Kit (No. PC0020, Solarbio, Beijing, China). Then, 100 µL of protein and 25 µL of SDS-PAGE Sample Loading Buffer (5X) (No. P0015, Beyotime, Shanghai, China) were mixed and added into a centrifugal tube, which was then boiled in a 98 °C metal bath for 10 min for denaturation. Then, 30 μg of proteins was separated by 10% SDS–PAGE (SF 10, Affinibody, Wuhan, China) and then electrotransferred to PVDF membrane (300 mA, 40 min). The PVDF membrane was rinsed with TBST (Solarbio, Beijing, China; Cat. No. T1083) for 15 min 3 times. Next, Y-Tec 5 min Ready-to-Use Blocking Buffer (No. YWB0501, Yoche, Shanghai, China) was incubated for 5 min at room temperature and incubated with primary antibodies (Bioss bs-1409R, 1:1500, Beijing, China; Affinity T0022, 1:800, Changzhou, China) overnight at 4 °C. After that, the PVDF was washed with TBST 3 times and incubated with HRP Conjugated secondary antibody at 37 °C for 1 h. ACTB (Bioss, 1:2000, Beijing, China) was used as a reference control. After that, a chemiluminescence detection system (iBright CL100, Thermo, Massachusetts, USA) was employed for quantitating the immunoreactive bands. The relative protein levels of HSF2 were analyzed as in a previous study [31].

### 2.7. Immunohistochemical Staining

IHC was carried out according to previous protocols with minor modifications [32]. For IHC analysis, 4 µm sections were deparaffinized and rehydrated in a graded series of ethanol to water, followed by hydration with an ethanol gradient (100–70%) as described in our previous study [33]. Then, they were antigen-repaired with 10 mM sodium citrate (pH 6.0). After cooling to room temperature, the sections were shaken in PBS and washed in 3% aqueous peroxide to block endogenous peroxide enzymes. Then, all sections were incubated with the primary HSF2 antibody (Bioss, bs-1409R, 1:300, Beijing, China) overnight at 4 °C. After washing with PBS 3 times, a biotinylated secondary antibody was incubated for 40 min, followed by 3,3′-diaminobenzidine (DAB) and hematoxylin staining. The negative control staining was captured by removing the primary HSF2 antibody. Images were obtained using a Zeiss LSM800 confocal microscope (Carl Zeiss, Oberkochen, Germany), with no less than 3 individuals and fields of view.

### 2.8. Bisulfite Sequencing PCR

The sequences of the CpG island in the yak and cattle-yak *HSF2* promoter region were treated as templates, and then MethPrimer (http://www.urogene.org/methprimer/ (accessed on 26 November 2023)) was used for designing primers (shown in Table 1) to target methylated and unmethylated sequences. Yak and cattle-yak DNA extractions were conducted according to the instructions of the Cells Genomic DNA Extraction Kit (No. D3396, Omega, Guangzhou, China). The quantity and quality of the extracted genomic DNA were determined by UV absorption. Then, the purified DNA was treated under the manufacturer’s instructions with the 2× EpiArt^®^ DNA Methylation Bisulfite Kit (No. EM202, Vazyme, Nanjing, China). Methylated cytosine became uracil after sulfite treatment (C-U conversion), otherwise unchanged. The reaction conditions are as follows: 93 °C for 6 min, followed by 42 cycles of 93 °C for 28 s, 58 °C for 28 s, 72 °C for 28 s, and then 72 °C for 6 min. The PCR products were detected by 1.5% agarose gel electrophoresis and then delivered to Sangon Biotech Co. (Shanghai, China) for sequencing with at least ten repeats.

### 2.9. Statistical Analysis

Statistical analyses were performed by GraphPad Prism8 (GraphPad, San Diego, CA, USA). A two-tailed Student’s *t*-test was utilized as an indicator of the significance of the comparisons shown. The multiple comparison tests were used to analyze *p*-values for more than two groups. The BSP sequencing results were analyzed by Vector NTI Advance 11.5.1. All data were presented as mean values ± standard error of the mean (SEM), and values of *p* < 0.05 were considered statistically significant. For multiple means comparisons, all statistical data were analyzed by one-way analysis of variance (ANOVA) using SPSS 19.0. The fold change in mRNA expression was calculated by using the mean expression values of three replicates from each group.

## 3. Results

### 3.1. Cloning of HSF2 Genes from Yak and Cattle-Yak and Physicochemical Property Analysis

Complementary DNA extracted from the yak and cattle-yak were employed as templates to amplify the CDS region of the *HSF2* gene via RT-PCR. The target bands were as expected according to the 1.5% gel electrophoresis assay. After sequencing and comparison, we obtained the sequence of the *HSF2* gene of the yak and cattle-yak containing 1617 bp and 1671 bp, respectively (Figure 1A). The CDS region of the yak *HSF2* gene was 1551 bp, encoding 516 amino acids, while that of the cattle-yak was 1605 bp and encoded 534 amino acids (Figure 1B). After prediction and comparison of encoding amino acids, we found that 54 bases were increased in the cattle-yak and additionally encoded 18 amino acids more than those of the yak from N-terminal 393 to 410 (Figure 1C). In addition, there was one base change from T to C in the cattle-yak compared to that of the yak, which was a missense mutation and turned lysine into glutamic acid.

The obtained amino acid sequence was applied to further evaluate the structural and physicochemical properties of the HSF2 protein in the cattle-yak. The predicted molecular weight of the cattle-yak HSF2 protein was 60.13 kDa, with a theoretical isoelectric point (pI) of 4.66. Interestingly, the ratio of the negatively charged residues (Asp + Glu) to positively charged residues (Arg + Lys) was 1.67 in the cattle-yak, which was similar to that of yak (1.64). Moreover, the instability index was computed to be 54.41, which defined HSF2 as an unstable protein. Furthermore, the prediction of the HSF2 protein domain indicated no signal peptide could be discovered, suggesting it cannot be transported and will be reserved in the synthesis region.

### 3.2. Molecular Characterization of Cattle-Yak HSF2 Protein

To investigate the molecular characterization of the HSF2 protein, the amino acid sequence of cattle-yak HSF2 and its counterparts from 12 mammalian species was aligned with the NCBI database. The analysis revealed that the HSF2 protein of the cattle-yak showed high identity with corresponding proteins from cattle (99.81%) and yak (97.63%), which indicated high conservatism during genetic evolution. In addition, the phylogenetic tree was constructed using MEGA7 software and also demonstrated that the cattle-yak HSF2 was clustered in the same isoform as cattle and yak (Figure 2), which is consistent with the ortholog analysis.

In order to explore the structure and function of the HSF2 protein, we predicted the secondary structure of the cattle-yak HSF2 protein using SOPMA software and showed a composition with 44.94% alpha helix, 9.93% extended strand, 4.31% beta-turn, and 40.82% random coil (Figure 3A). Furthermore, the tertiary structure of the cattle-yak HSF2 protein was found to be unstable (Figure 3B). According to the analysis of the interaction network, the predicted HSF2 protein of the cattle-yak primarily interacted with members of the heat stress and oxidative stress pathways (Figure 3C), such as HSF1, SIRT1, RBM44, and VASH2.

### 3.3. Expression Profile of HSF2 in Cattle-Yak Tissues

In order to explore the expression profile of *HSF2* mRNA in the cattle-yak, we compared the differential expression pattern of *HSF2* in different cattle-yak tissues using RT-PCR and RT-qPCR, including heart, liver, spleen, lung, kidney, intestine, muscle, and testes tissues. The data indicated that the transcript abundance of the *HSF2* gene was significantly higher in the testes than that in other tissues (Figure 4A,C). Thus, we hypothesized that the *HSF2* gene may be involved in regulating the development of testicular tissue in the cattle-yak and play a crucial role in controlling male reproduction. We further investigated the spatiotemporal expression of *HSF2* mRNA expression during testes development in the cattle-yak by RT-qPCR (Figure 4B,D). Our findings implied that the relative expression of *HSF2* mRNA in the testes of different ages in the cattle-yak tended to continuously increase along with sexual maturity. Specifically, there was no remarkable difference in the transcription levels of *HSF2* between fetal stage (6 months old) and juvenile (1 year old) cattle-yak (*p* > 0.05). However, the relative expression abundance was significantly higher in adult (3 years old) cattle-yak compared with other age groups (*p* < 0.05). Thus, the testes of adult cattle-yak were chosen as experimental subjects for the subsequent research.

### 3.4. HSF2 Expression in Cattle-Yak Testes Is Reduced Compared to Cattle and Yak

Considering the preceding results of *HSF2* expression abundance in the testes of cattle-yak, we attempted to test if the reproductive dysfunction of cattle-yak was correlated with the abnormal expression profile of the *HSF2* gene. To verify this hypothesis, the expression of mRNA patterns and protein levels of HSF2 in cattle-yak were compared with age-matched yak and cattle by RT-qPCR and Western blotting, respectively. According to the results of the PCR (Figure 5A) and RT-qPCR (Figure 5C), we found that the expression level of *HSF2* was significantly lower in adult cattle-yak than that of other stages (*p* < 0.05). Interestingly, there was a numerically lower amount of *HSF2* mRNA in cattle than in yak, but no remarkable statistical difference (*p* > 0.05). Consistent with expectations, the HSF2 protein levels in cattle and yak testes were predominantly increased compared to cattle-yak (Figure 5B,D, *p* < 0.05). Taken together, our data has been confirmed repeatedly that the differential expression of the *HSF2* gene in the cattle-yak may be the potential factor of male infertility, and the specific regulatory mechanism still needs further analysis.

### 3.5. Localization Analysis of HSF2 in Testes

Based on the significant difference in *HSF2* mRNA and protein expression in the testes of different groups, immunohistochemical staining was then applied to inspect the subcellular localization and protein levels of HSF2 in the testes of adult cattle, yak, and cattle-yak (Figure 6). It was shown that there was no significant difference in the morphology of seminiferous tubules in the testicular tissues of adult cattle and yak. However, compared with them, the seminiferous tubule of cattle-yak was markedly decreased in average diameter, accompanied by the cavity being enlarged. Moreover, the spermatogenic epithelium was dramatically wrinkled, resulting in a significant reduction in cellular composition and total cell number. Obviously, there were a few spermatocytes (PS), spermatogonia (SP), and Sertoli cells (SC) in the cattle-yak testes, which was a remarkable decrease compared with corresponding testes of cattle and yak. Contrary to expectations, it was almost impossible to find the secondary spermatocyte (SS) and sperm in cattle-yak testes. In order to confirm the subcellular localization and function of HSF2, the IHC staining noted that HSF2 was primarily located in the SP, PS, SS, and Leydig cell (IC), and the protein levels of HSF2 in the testes of cattle and yak were significantly higher than that of HSF2 in cattle-yak (Figure 6, *p* < 0.05).

Furthermore, we detected the localization and protein levels of HSF2 in cattle-yak among different stages to explore the dynamic changes in HSF2. HSF2 was detected at very low levels in the testes of fetal (5–6 months old) and juvenile (1–2 years old) cattle-yak, with slightly higher expression in adults (3–4 years old). We also found that a low number of spermatogenic cells were distributed throughout the testes, and spermatogenesis was mainly blocked at all ages (Figure 7, *p* < 0.05).

### 3.6. Hypermethylation of the HSF2 Promoter Region

Epigenetic modification is an important way to regulate gene expression. Thus, the BSP and sequencing were used to check the methylation status of the *HSF2* promotor region in the testes of yak, and cattle-yak. Thirty-five cytosine guanine (CpG) dinucleotides were predicted in the promotor regions of the *HSF2* loci in the three species. The target fragments were amplified and examined by PCR with the designed specific primers and agarose gel electropherograms (Figure 8B,C). After sequencing and analysis, we found that the methylation level of the cattle-yak had a significant increase compared to its yak counterparts (Figure 8D,E, *p* < 0.05). These researchers declared that the promotor region of *HSF2* was hypermethylated in the testes of cattle-yak.

## 4. Discussion

As a hybrid F1 generation of female yak and male cattle, the cattle-yak has significant heterosis in milk and meat production, but the male cattle-yak is infertile [34]. Although there is much research on the molecular mechanisms of male infertility in the cattle-yak, the specific underlying molecular mechanism remains obscure. A previous study on male fertility revealed that the *HSF2* gene was indispensable for regulating the spermatogenesis process in mice [35]. In this study, we first obtained the sequence of the *HSF2* gene in the cattle-yak and analyzed the biological characteristics of the cattle-yak. Our results showed that there was a missense mutation and a 54 bp extra fragment in the cattle-yak *HSF2* gene. Furthermore, the mRNA and protein levels of *HSF2* in cattle-yak testes were significantly lower than in their counterparts. Moreover, the promotor region of the *HSF2* gene in cattle-yak testes was hypomethylated. These data could be beneficial for understanding the physiological functions of *HSF2* in male reproduction and exploring the molecular mechanism of male cattle-yak sterility.

Spermatogenesis as well as its maturation are extremely complex biological processes that are regulated by multifactorial factors and intricate interaction networks. Recent evidence has suggested that heat shock proteins play a vital role in spermatogenesis and male reproduction when external environmental temperatures undergo drastic changes [36]. Likewise, previous reports have described that the abnormal expression or dysfunction of HSPA2, which was abundantly expressed in the testes, significantly affected spermatogenesis and sperm maturation [37]. This effect was reflected not only in the process of spermatogenesis, but also in the functional transformation of spermatozoa, which may ultimately lead to reduced sperm counts, abnormal morphology, and dysfunction of male reproduction [37]. In addition, growing evidence has demonstrated that HSFs are also closely related to spermatogenesis by regulating meiosis [14,38]. The total number of spermatozoa was significantly decreased with the knockout of the *HSF2* gene in mice, accompanied by a notable increase in the abnormal head morphology compared to that in normal male mice [37]. Therefore, we inferred that the *HSF2* gene may be associated with spermatogenesis and sperm maturation in the cattle-yak via meiosis by regulating the expression levels of the *HSF2* gene.

In this study, the *HSF2* gene was cloned, and the expression patterns were analyzed in the yak, cattle, and the cattle-yak. There was a novel splice site in the *HSF2* gene in the cattle-yak that introduced an additional 54 bp that encode 18 amino acids, which could cause changes in protein activity and function of the *HSF2* gene. Additionally, a missense mutation was also found (T to C) that caused lysine acid to become glutamic acid in the *HSF2* CDS region. Similarly, a previous study revealed that in humans, the R502H mutation of *HSF2* would contribute to the onset of male fertility potential and the process of meiosis of spermatogenic cells, leading to idiopathic azoospermia (IA) in males [26]. Likewise, it was also reported that mutations in the *HSF2* gene were associated with non-obstructive azoospermia (NOA) in humans [39]. However, genotyping studies, along with the *HSF1* and *HSF2* functional assays carried out in 2021, indicated that the rare genotype frequency was lower than another possibility [40]. Thus, although previous studies have demonstrated that the *HSF1* and *HSF2* genes are indispensable for spermatogenesis, their functional insights into genotype sequencing and male infertility are still inconsistent. Moreover, our study inferred that these sequence changes of *HSF2* may influence the function of HSPs and lead to the anomalous expression of HSPs, which may be associated with male sterility in the cattle-yak. Admittedly, whether these differences between cattle-yak and yak HSF2 proteins alter protein function and biological function needs further investigation.

Additionally, the ortholog analysis indicated the *HSF2* in cattle-yak testes was highly conserved with other species during genetic evolution and showed that the *HSF2* in cattle-yak had a similar homology with their parents. Remarkably, the members of the HSF family were interrelated and formed a massive network regulatory effect that synergistically regulated heat stress under specific conditions. In the present study, the HSF2 protein potentially displays close interactions with numerous proteins, including HSF1, SIRT1 that deacetylates transcription factors, the RBM44 germ cell intercellular bridge protein, and VASH2 that mediates microtubule detyrosination. These findings are in agreement with previous observations that HSF2 and HSF1 form heterotrimer complexes and activate essential genes in specific conditions [41]. However, the proteins and mechanisms that produce reciprocal effects with HSF1 and HSF2 proteins remain scarce because both are involved in spermatid differentiation. Previous reports have described that the deficiency of *HSF2* may not only affect its function but also damage the formation of *HSF1* [39]. By coincidence, *HSF2* has been reported to be essential for the cell cycle regulation process through cooperation with subfamily member *HSF1* [27,42]. Meiosis, the unique cell proliferation and division mode of germ cells, is an important process in sexual reproduction, and the disorder of meiosis may fail to generate normal generative cells and embryos [14,38]. Both *HSF1* and *HSF2* are involved in the regulation of meiosis in spermatogenesis and oogenesis, which can form heterotrimers upon stress to protect reproductive capability from environmental changes [43]. Amazingly, we found that HSF2-encoded proteins were remarkably different in yak and cattle. To further clarify the potential reasons for these discrepancies, their structures and interacting proteins were predicted using online software, and it was found that these potential proteins interacting with cattle-yak *HSF2* were almost playing different roles in mitosis and meiosis, which could provide a novel insight into the next step in resolving the molecular mechanism of *HSF2* in male cattle-yak sterility.

The testes, specific tissues in males, are a known cofactor for producing and preserving sperm and secreting androgens, and all these physiological processes cannot be separated from the ordered expression of related genes. Herein, we confirmed that *HSF2* mRNA was ubiquitously expressed in various tissues and was abundantly expressed in cattle-yak testes, which was typically observed in mammals and consistent with other previous studies. The *HSF2* gene is primarily expressed in the testes of mice, humans, sheep, and other species [23,26,44]. Androgens and androgen receptors are essential for spermatogenesis; in addition, Leydig cells are required for androgenesis and testicular development. The present study pointed out that the decreased expression of the AR (androgen receptor) in adult cattle-yak may prevent spermatogenesis [45], which suggests that the decreased expression of the *HSF2* gene in adult cattle-yak testes may be related to abnormal hormone regulation, thus leading to blocked spermatogenesis. However, the association between *HSF2* expression and androgens needs to be further investigated. It was demonstrated that *HSF2* gene-deficient male mice exhibited an increase in the vacuolization of the varicocele and an increase in apoptotic cells through disrupting the meiosis process in the spermatogonia and spermatogenesis [23], which was consistent with our findings in the cattle-yak. Equally, the HSF2 protein was found to be localized in sperm nuclei in mice, accompanied by the HSF1 protein [46]. Likewise, the localization and expression profiles of the HSF2 protein in cattle-yak testes indicated that meiosis in male cattle-yak was blocked at the primary spermatogonia in the seminiferous tubule, with few SCs, PS, and SP, which was in agreement with the previous report [47]. However, Sertoli cells play direct or indirect roles in the regulation of spermatogenesis by providing protection and nutrition for developing spermatozoa. A decrease in Sertoli cells may be associated with the failure of spermatogenesis, but the specific regulatory mechanisms still need to be further investigated. Since HSF2 is involved in the regulation of meiosis, we proposed that the defects of *HSF2* on the primary spermatocytes in cattle-yak are one of the main reasons for the failure of meiosis.

DNA methylation, as an important epigenetic modification, is associated with mammalian gametogenesis and embryonic development through changing the activity of the related genes [48,49]. In mammals, DNA methylation has been shown to control mRNA abundance and regulate transcriptional activity in vivo, which upregulates or downregulates the interaction pathways [50]. DNA methylation occurring during spermatogenesis can lead to the aberrant expression of the relevant genes, which may cause male infertility [51]. Furthermore, the available data suggest that the *HSF2* gene accounts for spermatogenesis by regulating the cell cycle in human; thus, the misregulation of the *HSF2* gene may cause an increased risk of idiopathic azoospermia (IA) [26]. Similarly, anomalous spermatogenesis occurred after the knockout of the *HSF2* gene in mice, thereby causing reduced testes size and sperm count [24]. These studies have inspired the idea that the decrease in the expression level of the *HSF2* gene may be associated with male fertility. These lead us to hypothesize that these misregulations of the *HSF2* gene were attributed to DNA methylation. To confirm this assumption, we compared the methylation status of the *HSF2* promotor region in cattle-yak and yak testes. As per our expectation, the CpGs in cattle-yak were hypermethylated, which was an underlying cause for *HSF2* mRNA differential expression between them. Therefore, we suggested that the *HSF2* hypermethylation of the promoter region in cattle-yak leads to the suppression of promoter activity, which suppresses *HSF2* gene expression and transcription. Previous studies into aberrant gene expression in cattle-yak revealed that appropriate DNA methylation and histone modifications are vital for testicular tissue development and spermatogenesis [52,53]. It was reported that higher DAN methylation and the aberrant expression of AcK9 may contribute to disrupting the development of cattle-yak germ cells [52], which was in accordance with our finding that the hypermethylation of the *HSF2* promoter region may be one of the underlying cofactors in destroying the reproduction of male cattle-yak. Furthermore, findings of histone methylations revealed the depletion of *H3K4* in cattle-yak testes compared to yak, which suggested that histone methylation regulation plays an important role during spermatogenesis [53]. Overall, our data first noted that the attenuated expression of *HSF2* in cattle-yak testes may be the latent origin of meiotic arrest, which could induce male infertility in cattle-yak. Although the differential expression and epigenetic modification of the *HSF1* gene in the same family cluster were not investigated, the interaction of *HSF2* with *HSF1* may involve collaborative regulation of spermatogenesis in male cattle-yak, which needs more experiments to verify.

Taken together, despite its highly conserved evolution and enriched expression in the testes, the genetic structure and biological characteristics of the cattle-yak *HSF2* gene were remarkably different between cattle and yak. The low expression and hypermethylation of *HSF2* could be potential causes of the obstruction of spermatogenesis, which leads to male cattle-yak infertility. Certainly, the underlying molecular mechanisms of male cattle-yak infertility were intricate and complex, thus requiring integrative analysis from multiple perspectives and omics for comprehensive elucidation. Notwithstanding the limitations, our study provided the basic guidelines for the *HSF2* gene in male reproduction and provided a conceptual framework for the biological roles of *HSF2* in male cattle-yak sterility.

## 5. Conclusions

In brief, although the *HSF2* gene was predominantly expressed in male cattle-yak testes, its expression level was significantly lower than that of its parental counterparts. Apart from these, methylation levels were confirmed to be significantly increased, which indicated that the decrease in *HSF2* may be attributed to DNA methylation, thereby causing reproductive dysfunction in male cattle-yak. These sites have provided a conceptual framework for exploring the underlying mechanisms of sterility in male cattle-yak.

## Figures and Tables

**Figure 1 animals-14-01410-f001:**
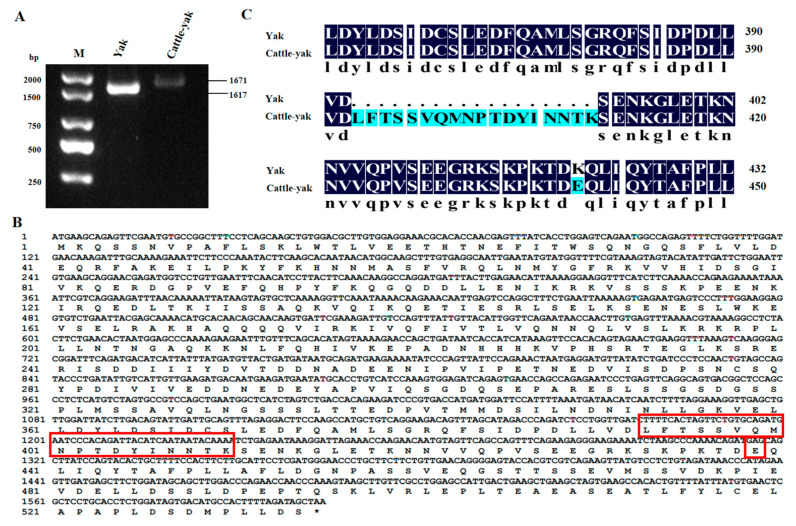
The PCR amplification results of the *HSF2* gene. (**A**) The agarose gel electrophoresis of yak and cattle-yak *HSF2* gene cloning products, M: DNA maker (2000 bp). (**B**) The nucleotide sequence of the CDS region in the cattle-yak *HSF2* gene and the predicted amino acid sequence of the cattle-yak HSF2 protein. The red boxes represent the additionally encoded sequence of 54 bp in cattle-yak and a missense mutation. (**C**) The comparison of amino acid sequences between the yak and cattle-yak.

**Figure 2 animals-14-01410-f002:**
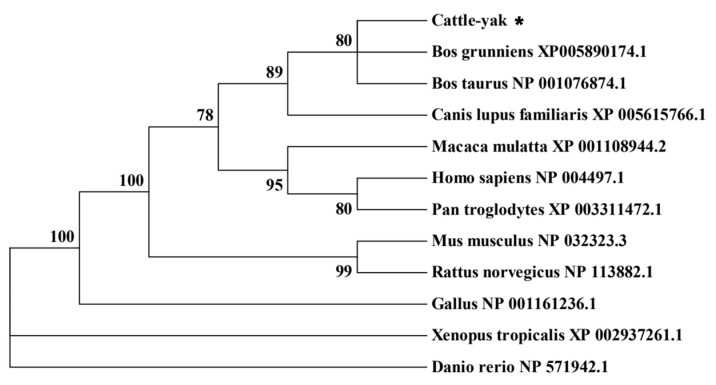
The phylogenetic tree of *HSF2* amino acid sequences. The notation “*” represents the cattle-yak.

**Figure 3 animals-14-01410-f003:**
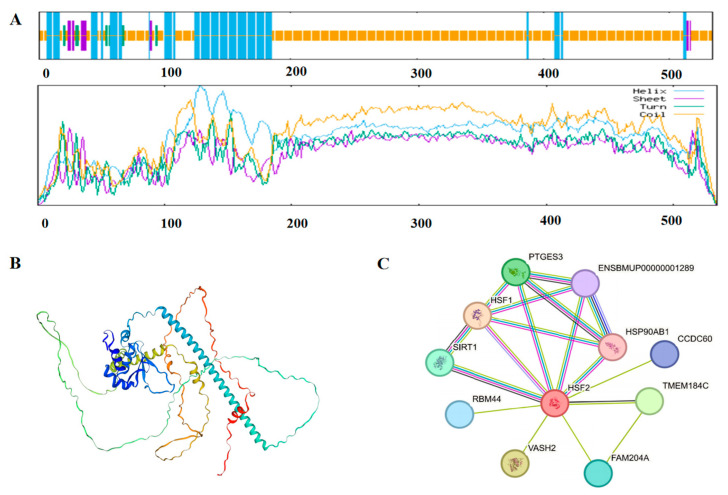
The predicted spatial structure and interaction proteins of the cattle-yak HSF2 protein. (**A**) The predicted secondary structure of the cattle-yak HSF2 protein. Blue line: alpha helix; Yellow line: random coil; Purple line: extended strand; Green line: beta-turn (**B**) The predicted tertiary structure of the cattle-yak HSF2 protein. (**C**) Interaction network for the cattle-yak HSF2 protein.

**Figure 4 animals-14-01410-f004:**
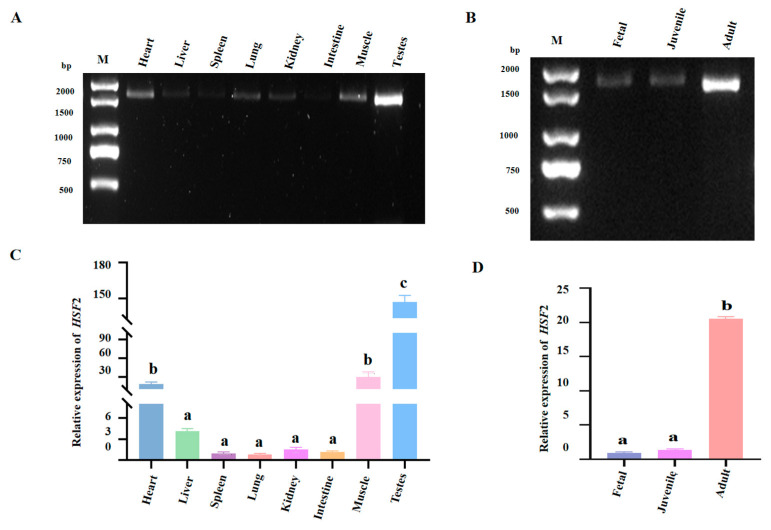
The expression profile of *HSF2* in the cattle-yak. (**A**) The PCR products of *HSF2* in different tissues of adult cattle-yak were detected by agarose gel electrophoresis. (**B**) The PCR amplifications of *HSF2* gene in the testes of cattle-yak from different growth stages were detected by agarose gel electrophoresis. Fetal: 5–6 months old, juvenile: 1–2 years old, adult: 3–4 years old. (**C**) The RT-qPCR analysis of the *HSF2* gene in different cattle-yak tissues. (**D**) The expression levels of the *HSF2* gene in the testes of different cattle-yak growth stages. Different letters indicate significant differences (*p* < 0.05). The same letter means that the difference is not significant.

**Figure 5 animals-14-01410-f005:**
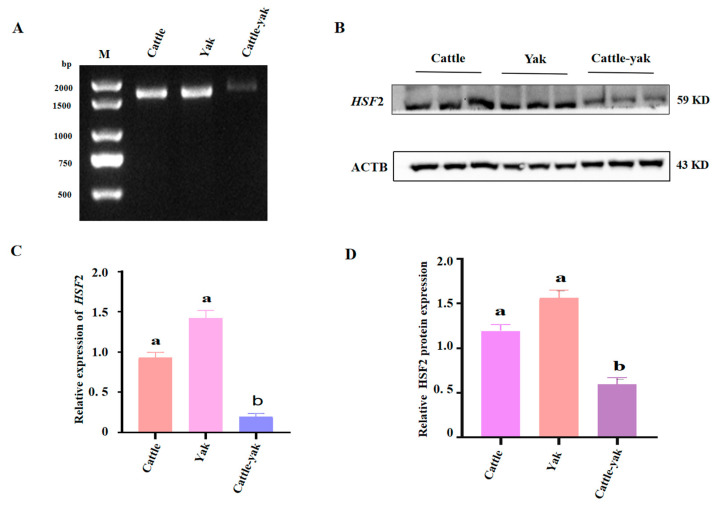
The mRNA and protein expression levels of *HSF2* in cattle, yak, and cattle-yak. (**A**) The differential expression of *HSF2* mRNA in the testes of cattle, yak, and cattle-yak by agarose gel electrophoresis. (**B**) The expression levels of the HSF2 protein in the testes of cattle, yak, and cattle-yak via Western blot; ACTB served as a reference control. (**C**) The relative *HSF2* mRNA expression levels among cattle, yak, and cattle-yak via RT-qPCR. (**D**) Quantization showed the expression of HSF2 protein in the testes from cattle, yak, and cattle-yak. Relative changes in HSF2 protein in the testes were presented after normalization with ACTB. Different letters indicate significant differences (*p* < 0.05). The same letter means that the difference is not significant.

**Figure 6 animals-14-01410-f006:**
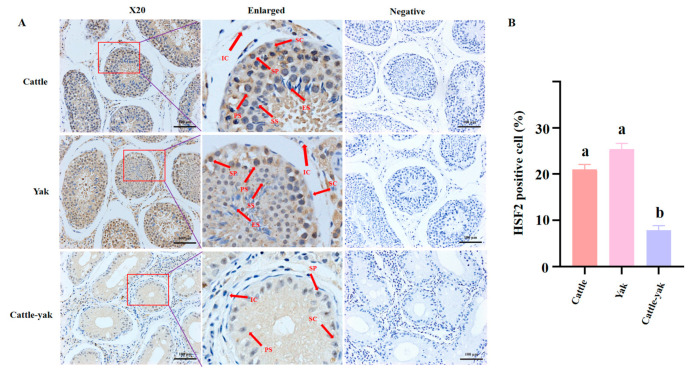
(**A**) The localization and expression of HSF2 in adult (3–4 years old) cattle, yak, and cattle-yak. All testes were embedded in paraffin and cut into 4 μm thickness for staining with the anti-HSF2 primary antibody (brown). Hematoxylin was counterstained to label the nucleus (blue). The red box was enlarged to include the following images: SP: spermatogonia; PS: primary spermatocyte; SS: secondary spermatocyte; SC: Sertoli cell; IC: Leydig cell; ES: elongating spermatid. The scale bar of the enlarged images was 100 μm. (**B**) The relative HSF2-positive cell of the testes in cattle, yak, and cattle-yak (*p* < 0.05). The same letter means that the difference is not significant.

**Figure 7 animals-14-01410-f007:**
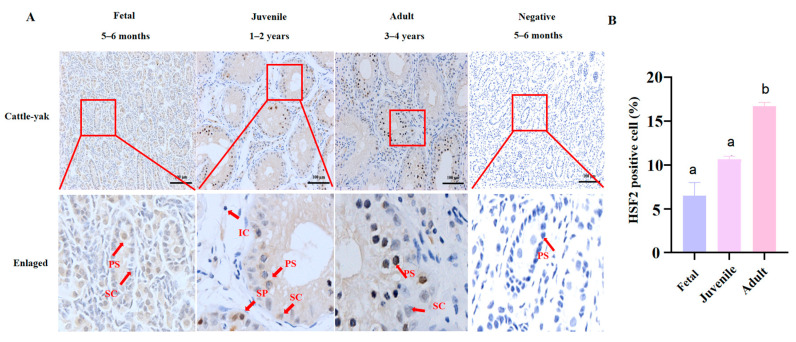
(**A**) The localization and expression of HSF2 in fetal (5–6 months old), juvenile (1–2 years old), and adult (3–4 years old) cattle-yak. All testes were embedded in paraffin and cut into 4 μm thickness for staining with the anti-HSF2 primary antibody (brown). Hematoxylin was counterstained to label the nucleus (blue). The red box was enlarged to the following images. SP: spermatogonia; PS: primary spermatocyte; SC: Sertoli cell; IC: Leydig cell; ES: elongating spermatid. The scale bar of enlarged images was 100 μm. (**B**) The relative HSF2-positive cell of the testes in cattle-yak from different stages (*p* < 0.05). The same letter means that the difference is not significant.

**Figure 8 animals-14-01410-f008:**
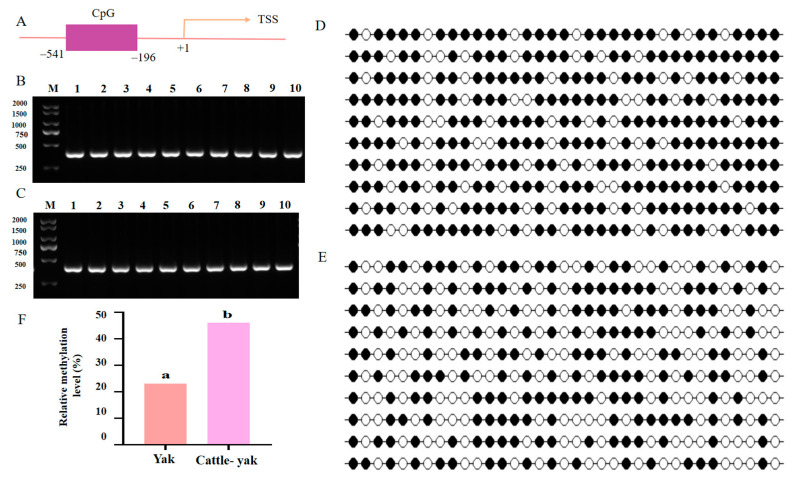
DNA methylation analysis of *HSF2* promoter regions in the testes of yak and cattle-yak by bisulfite sequencing. (**A**) The schematic indicates the position of the analyzed CpG islands in the promoter region of the *HSF2* gene. (**B**,**C**) The BSP-amplified products of the bisulfite-converted DNA sequence of the *HSF2* promotor region in yak (**B**) and cattle-yak (**C**) testes, with ten replicates, respectively. (**D**,**E**) The methylation status of CPG sites, the black circles represent unmethylated CpGs, and the white circles represent methylated CpGs of yak (**D**) and cattle-yak (**E**). (**F**) Percentages of methylated CpG sites in yak and cattle-yak testes calculated from (**D**,**E**). The different superscript letters showed significant differences (*p* < 0.05). The same letter means that the difference is not significant.

**Table 1 animals-14-01410-t001:** PCR primer sequences of *HSF2*.

Primer Name	Primer Sequence (5′-3′)	Product Size (bp)	Annealing Temperature (°C)	Utilization
*HSF2*	F_1_:GCGTTTGGGTGTAGAATCTGG R_1_: ACGTAGCGTCCACTTCTTG	1671 bp	65 °C	RT-PCR
*HSF2*-Q	F_2_:TGCAGATGAATCCCACAGATTACA R_2_:GCTGCTTATCTGTTTTGGGC	129 bp	60 °C	RT-qPCR
*β-actin*	F_3_:GATGATATTTGCTGCGCTCGTG R_3_:CTTGCTCTGAGCCTCATCCC	177 bp	60 °C	RT-qPCR
*HSF2*-B	F_4_:AAAGGTTTTTTTACGTGAAATTATT R_4_:AAAATTCCAAATTCTACACCCA	346 bp	48 °C	BSP

## Data Availability

Data sources are included in this article.

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
