# Peer review of "The Expression and Epigenetic Characteristics of the HSF2 Gene in Cattle-Yak and the Correlation with Its Male Sterility"

_animals, 2024, doi:10.3390/ani14101410_

Round 1

Reviewer 1 Report

Comments and Suggestions for Authors

In the study, the authors studied the sequence of cattle-yak HSF2 gene and its expression in in testis tissues by comparing with those of yak. And they identified the mutations of the gene and found that its low expression and high methylation level in cattle-yak is related with the male cattle-yak sterility. The findings of the study are helpful for understanding the mechanism of male cattle-yak sterility.

The following points should be clarified by the authors before the manuscript could be accepted.

1.     For the HSF2 mutations found in cattle-yak, the authors should make it clear whether they are cattle-yak hybrid-specific mutations. In my opinion, they should sequence or genotype a number of cattle, yaks, and their hybrids to investigate the frequencies of the mutations.

2.     How about the methylation level of HSF2 gene in testis tissue of cattle? Is it also lower than that of cattle-yak hybrid?

3.     In the Materials and Methods part, the authors should provide the necessary information on samples of yak and cattle, such as the sample number, breed, and the sampling locations.

4.     The authors should have a thorough check of the writing of their manuscript. There are some obvious writing errors, such as,

Line 66, “I addition” should be changed to “In addition”

Line 180, “30 ug” should be changed to “30 μg”

Line 366, For the samples “the testis of cattle, yak, and cattle-yak”, “cattle” should be deleted as there is no cattle results in this section.

Line 507, “pro-vides should be changed to “provide”

Comments on the Quality of English Language

The authors should carefully check the English writing of their manuscript.

Author Response

Dear reviewer,

Thank you very much for taking your time to review this manuscript. We really appreciate all your comments and suggestions. Please find my response in below. We hope that the revised manuscript is accepted for publication.

Best wishes,

Sincerely yours,

Qinhui Yang,

2024/4/27

Comments:

  1. For the HSF2 mutations found in cattle-yak, the authors should make it clear whether they are cattle-yak hybrid-specific mutations. In my opinion, they should sequence or genotype a number of cattle, yaks, and their hybrids to investigate the frequencies of the mutations.

Response: Thanks for your comments. Because the sequence of cattle could be found from ncbi, we then sequenced and analyzed yak and cattle-yak. However, the regulation of testicular function is complex. In this study, we foused on the expression pattern of HSF2 gene in cattle-yak and tried to established the methylation pattern of HSF2 promoter region by BSP method, then attempted to reveal the correlation between HSF2 and the male cattle-yak sterility.

  1. How about the methylation level of HSF2 gene in testis tissue of cattle? Is it also lower than that of cattle-yak hybrid?

Response: Thanks for your professional review. We apologized that we didn’t sequence the methylation level of HSF2 gene in testis tissus of cattle. But, according to previous work, we discovered that there was no significant difference between cattle and yak. The causes of male cattle-yak infertility were various andcomplex. Besides, we took the environmental characteristics into account, then decided to choose yak which was living in Qinghai-Tibetan Plateau as cattle-yak to be the comparison object. Thanks for your comments again.

  1. In the Materials and Methods part, the authors should provide the necessary information on samples of yak and cattle, such as the sample number, breed, and the sampling locations.

Response: Thanks for your suggestion. We sincerely appreciate the valuable comments. We have checked the Materials and Method (M-M) section, and apologized that the number of each evaluated group was ambiguous. We have improved this part which could be found from the revised version: “ Each tissue sample, including heart, liver, spleen, lungs, kidneys, intestine, muscles, and testis, was captured from three adult yaks and cattle-yaks respectively in a pasture in Hongyuan County, Sichuan Province (Altitude 3500 meters; longitudinal 102°44'E, latitudinal 32°41'N). Besides, the same tissue samples of adult cattle were obtained randomly from three healthy cattle in Qingbaijiang District, Sichuan Province. Herein, testis tissue samples in cattle-yak of fetal (6 months old), juvenile (1 year old), and adult cattle-yaks (3 years old) were obtained with three biological replicates, respectively. All samples were cleaned with autoclaved PBS, and then parts of them were put into cryopreservation tubes and brought back to the laboratory in liquid nitrogen tanks for subsequent experiments. The other parts of the tissues were fixed with 4% paraformaldehyde for IHC detection. Remarkably, all animals on this ranch lived in the same environment and enjoyed comparable nutritional conditions. In addition, the selected samples were all from individuals with healthy, similar age and body size. Each evaluated group involved three healthy individuals which were of the same age, similar physiological status and body size, and comprised 4 technical replicates for each trial.”

  1. The authors should have a thorough check of the writing of their manuscript. There are some obvious writing errors, such as,

Line 66, “I addition” should be changed to “In addition”

Line 180, “30 ug” should be changed to “30 μg”

Line 366, For the samples “the testis of cattle, yak, and cattle-yak”, “cattle” should be deleted as there is no cattle results in this section.

Line 507, “pro-vides“ should be changed to “provide”

Response: Thanks for your valuable suggestions. We are sorry for these mistake, and have improved them. Thanks for your efforts and providing your valuable feedback again. Please check the revised version:

(1) In addition, the changes in environment such as temperature, oxygen con- centration, and ultraviolet may lead to cellular oxidative stress or epigenetic dysregulation, which were implicated in male infertility.

(2) Then 30 μg proteins were separated by 10% SDS–PAGE (SF 10, Affinibody, Wuhan, China) and then electrotransferred to PVDF membrane (300mA, 40min).

(3) Thus, the BSP and sequencing were used to check the methylation status of the HSF2 promotor region in the testis of yak, and cattle-yak.

(4) Notwithstanding the limitations, our study provided the basic guidance of HSF2 gene in male reproduction and provides a conceptual framework of biological roles of HSF2 in male cattle-yak sterility.

Reviewer 2 Report

Comments and Suggestions for Authors

Basically, the experimental design is well organized. However, the reviewer recommends the author further revision for publish this work.

Especially for the Fig. 6 immunohistochemistry data. Compared to the Figure 7, Cattle and Yak immuno-staining are not clear because of high background. Although HSF2 should localized in nuclear, in Fig.6 cytoplasmic stain are appeared in cattle and Yak. Furthermore, Fig. 6B, should describe about the HSF2 positive cell % instead of positive area because the staining should be in nuclear not in cytoplasm. Furthermore, authors should add the information in materials and methods about the negative control staining for IHC and internal control for western blotting. Furthermore, the information about the season to collect each sample should be cited, because depending on the timing, maturation rate will be different in Yak and cattle-yak.

Discussion is unclear. Authors should carefully summarize and discuss. For example, line 469-473, are there any relationship between HSF2 expression and Sertoli cells? Authors should discuss about the somatic cell roles for spermatogenesis to refer after showing the results.  For example, Leydig cells in yak- cattle (Sato et al., Reproduction in Domestic Animals, 2020). In addition, to control the gene expression, epigenetic modulation is important. Please discuss it with proper references, for example, Phakdeedindan et al. (Reproduction in Domestic Animals, 2020) and Li et al. (Theriogenology, 2020). Furthermore, if the authors would like to discuss the collaboration of HSF1 and HSF2, the authors should add the data of HSF1 expression in cattle-yak. 

Comments on the Quality of English Language

There are several mis spelling of the words, such as " I addition " should be replaced into " In addition,", in vivo or in vitro should be written in Italic, "scar bars" should be replaced into " scale bars" and so on. Authors should use use the correct word in all manuscript.  

Author Response

Dear reviewer,

Thanks for your professional review work on our manuscript. As you are concerned, there are several problems that need to be addressed. According to your suggestions, we have made extensive corrections to our previous draft. Please check our following response to reviewers’ comments. We hope that the modified version is now suitable for publication.

Best wishes,

Sincerely yours,

Qinhui Yang

27/4/2024

Comments:

  1. Basically, the experimental design is well organized. However, the reviewer recommends the author further revision for publish this work.
  • Especially for the Fig. 6 immunohistochemistry data. Compared to the Figure 7, Cattle and Yak immuno-staining are not clear because of high background. Although HSF2 should localized in nuclear, in Fig.6 cytoplasmic stain are appeared in cattle and Yak. Furthermore, Fig. 6B, should describe about the HSF2 positive cell % instead of positive area because the staining should be in nuclear not in cytoplasm.

Response: Thanks for your comments and suggestions. The high background maybe related to excessive antibody concentration, but we analyzed the positive cells by the help of images, which could eliminate this influence and analyze the positive cells only (Fig.6B). Thanks for your suggestions again, we realize that the describe of Fig6.B was inaccurate. We have changed the “HSF2 positive area(%)” into “HSF2 positive cell (%)” . The description is even better.

  • Furthermore, authors should add the information in materials and methods about the negative control staining for IHC and internal control for western blotting. Furthermore, the information about the season to collect each sample should be cited, because depending on the timing, maturation rate will be different in Yak and cattle-yak.

Response: Thanks for your constructive suggestions. “The negative control staining was captured by removing the primary HSF2 antibody.” was added to describe the negative control of IHC in Materials and Methods.

“ACTB (Bioss, 1:2000) was served as a reference control.” was added to improve the information of western blotting in materials and methods.

And in the Sample Collection parts, “In this study, all the samples were collected in September, and they are in a similar physiological state.” was added to describe the season of sample collection.

  1. Discussion is unclear. Authors should carefully summarize and discuss. For example, line 469-473, are there any relationship between HSF2 expression and Sertoli cells? Authors should discuss about the somatic cell roles for spermatogenesis to refer after showing the results.  For example, Leydig cells in yak- cattle (Sato et al., Reproduction in Domestic Animals, 2020). In addition, to control the gene expression, epigenetic modulation is important. Please discuss it with proper references, for example, Phakdeedindan et al. (Reproduction in Domestic Animals, 2020) and Li et al. (Theriogenology, 2020). Furthermore, if the authors would like to discuss the collaboration of HSF1 and HSF2, the authors should add the data of HSF1 expression in cattle-yak. 

Response: Thank you for your insightful comment and constructive feedback. We have tried our best to improve the “Discussion part”. We hope that the improved version is now suitable for publication.

“Androgens and androgen receptors are essential for spermatogenesis, besides, Leydig Cell is required for androgenesis and testicular development. The present study pointed that the decrease expression of AR (androgen receptor) in adult cattle-yak may prevent the spermatogenesis [45], which suggest that the decreased expression of HSF2 gene in adult cattle-yak testis may be related to abnormal hormone regulation, thus leading to blocked spermatogenesis. However, the association between HSF2 expression and androgens needs to be further investigated.” was added to line 472-478.

“ Previous studies into aberrant gene expression in cattle-yak revealed that appropriate DNA methylation and histone modifications are vital for testicular tissue development and spermatogenesis [52,53]. It was reported that higher DAN methylation and the aberrant expression of AcK9 may contribute to disrupting the development of cattle-yak germ cells [52], which was in accord with our finding that the hypermethylation of the HSF2 Promoter Region may be one of the underlying cofactor to destroy the reproduction of male cattle-yak. Furthermore, finding of histone methylations discovered depletion of H3K4 in cattle-yak testis compared to yak, which suggested that histone methylation regulation plays important role during spermatogenesis [53].” was added to line 511-519.

Thanks for your kind suggestion. We have checked the literature and added more references to support this idea as your suggested. We feel great thanks for your professional review work on our article. We hope that the revised version is now suitable for publication.

[45]  Sato, Y., Kuriwaki, R., Hagino, S., Shimazaki, M., Sambuu, R., Hirata, M., Tanihara, F., Takagi, M., Taniguchi, M., Otoi, T. Abnormal functions of Leydig cells in crossbred cattle-yak showing infertility. Reprod. Domest. Anim. 2020, 55(2), 209–216.

[52]  Phakdeedindan, P., Wittayarat, M., Tharasanit, T., Techakumphu, M., Shimazaki, M., Sambuu, R., Hirata, M., Tanihara, F., Taniguchi, M., Otoi, T., Sato, Y. Aberrant levels of DNA  methylation and H3K9 acetylation in the testicular cells of crossbred cattle-yak showing infertility. Reprod. Domest. Anim. 2022, 57(3), 304–313.

[53]  Li, Y. C., Wang, G. W., Xu, S. R., Zhang, X. N., Yang, Q. E. The expression of histone methyltransferases and distribution of selected histone methylations in testes of yak and cattle-yak hybrid. Theriogenology. 2020144, 164–173.

  1. There are several mis spelling of the words, such as " I addition " should be replaced into " In addition,", in vivo or in vitro should be written in Italic, "scar bars" should be replaced into " scale bars" and so on. Authors should use the correct word in all manuscript.  

Response:Thank you for pointing out these problems. We have corrected the spelling, which could be checked in the revised version. Besides, we have checked our manuscript carefully in order to avoid these mistake. Thank you for your kind suggestion.